# Assembly of the Complete Mitochondrial Genome of *Pereskia aculeata* Revealed That Two Pairs of Repetitive Elements Mediated the Recombination of the Genome

**DOI:** 10.3390/ijms24098366

**Published:** 2023-05-06

**Authors:** Xue Zhang, Yuanyu Shan, Jingling Li, Qiulin Qin, Jie Yu, Hongping Deng

**Affiliations:** 1College of Horticulture and Landscape Architecture, Southwest University, Chongqing 400716, China; 2Key Laboratory of Agricultural Biosafety and Green Production of Upper Yangtze River, Ministry of Education, Southwest University, Chongqing 400715, China; 3Center for Biodiversity Conservation and Utilization, School of Life Sciences, Southwest University, Chongqing 400715, China

**Keywords:** mitochondrial plastid DNAs, mitogenome, *Pereskia aculeata*, recombination, repetitive elements, RNA editing

## Abstract

*Pereskia aculeata* is a potential new crop species that has both food and medicinal (antinociceptive activity) properties. However, comprehensive genomic research on *P. aculeata* is still lacking, particularly concerning its organelle genome. In this study, *P. aculeata* was studied to sequence the mitochondrial genome (mitogenome) and to ascertain the assembly, informational content, and developmental expression of the mitogenome. The findings revealed that the mitogenome of *P. aculeata* is circular and measures 515,187 bp in length with a GC content of 44.05%. It contains 52 unique genes, including 33 protein-coding genes, 19 tRNA genes, and three rRNA genes. Additionally, the mitogenome analysis identified 165 SSRs, primarily consisting of tetra-nucleotides, and 421 pairs of dispersed repeats with lengths greater than or equal to 30, which were mainly forward repeats. Based on long reads and PCR experiments, we confirmed that two pairs of long-fragment repetitive elements were highly involved with the mitogenome recombination process. Furthermore, there were 38 homologous fragments detected between the mitogenome and chloroplast genome, and the longest fragment was 3962 bp. This is the first report on the mitogenome in the family Cactaceae. The decoding of the mitogenome of *P. aculeata* will provide important genetic materials for phylogenetic studies of Cactaceae and promote the utilization of species germplasm resources.

## 1. Introduction

*Pereskia aculeata* Mill. is a climbing shrub plant [1] that belongs to the family Cactaceae. This species is native to South America and mainly in southern to northeastern Brazil [2]. Its leaves have a high nutritional value and are eaten by local people as a vegetable, known as “Barbados Gooseberry” [3]. For instance, the content of protein in the leaves of *P. aculeata* is much higher than that of other common vegetables [4]. In many economically underdeveloped areas, people often rely on it as the main source of protein in everyday life, so *P. aculeata* is also known as the “meat of the poor”. Moreover, in addition to minerals such as calcium, magnesium, manganese, and zinc, the leaves are also high in total dietary fiber, as well as vitamin A, vitamin C, and folate [5]. Apart from this, *P. aculeata* also has medical value. The leaves of *P. aculeata* are very high in mucus, which is often used in traditional Brazilian medicine to heal skin wounds and treat inflammation [6]. Furthermore, the functional mucilage extracted from *P*. *aculeata* has been shown to be rich in polysaccharides and proteins and can be used as an emulsifier and stabilizer in the food industry [7,8,9], also used as a corrosion inhibitor of steel [10].

The mitochondrion is a semi-autonomous eukaryotic organelle that has a small genome (mitogenome) that interacts with the nuclear and chloroplast genome to provide the biochemical machinery for energy conversion. Mitochondria synthesize Adenosine Triphosphate (ATP) through the tricarboxylic acid cycle and oxidative phosphorylation to provide energy for cells [11,12]. Mitochondria are also involved in cell differentiation, information transmission, and apoptosis [13]. Studies have shown that mitochondria are involved in maternally inherited cytoplasmic male sterility (CMS) [14]. Modern plant breeders focus on the process of rearranging the mitogenome to restore fertility in plants [15].

The plant mitogenome is more structurally complex and difficult to sequence compared to the chloroplast genome and is still in a booming stage of development [16]. By January 2023, The National Center for Biotechnology Information (NCBI) has released 9537 chloroplast genomes, 562 mitogenomes, and 1279 plastid genomes (https://www.ncbi.nlm.nih.gov/genome/browse#!/organelles/; accessed on 12 January 2023). Although the mitogenome of plants generally shows a circular genome structure [17], the physical organization of the mitogenomes includes a set of sub-genomic forms resulting from homologous recombination between repeats [18], with a mixture of linear, circular and branched structures [19]. For all organisms, even viruses, recombination is an important process in DNA replication [20,21,22] because recombination activities are involved in the rescuing of stalled or damaged replication forks. The recombination mechanism can be divided into two pathways: homologous recombination (HR) and non-homologous recombination [23]. HR is the main pathway of plant mitochondrial DNA repair [24], and it relies on the identification of sequences with high similarity to the region to be repaired. Non-homologous recombination pathways utilize very limited or non-sequence similarities for the repair of broken DNA. It mainly occurs in eukaryotes, which can lead to deletions or the duplications of sequences [25,26].

It is important to note that the regulation of HR in plant mitochondria is crucial to avoiding the occurrence of deleterious genomic rearrangements. Factors involved in HR or its regulation are sensitive and should be well understood to prevent instability and damage to the mitochondria. The endosymbiosis theory postulates that mitochondria arise from an endosymbiotic alpha-proteobacterium within an archaeal-derived host cell and finally evolved into organelles of eukaryotic cells [27]. Thus, the plant mitochondrial HR is similar to bacterial HR [28,29,30]. It has been suggested that mitochondrial HR is a frequent and reversible recombination involving large repeats. If these changes do not harm the function of the mitochondria, they will be retained, resulting in an overall increase in mitogenome size [31].

Although nearly 500 plant mitogenomes have been reported since 1990, there are no mitogenome studies of plants from the cactus family. Here, we completed the assembly and annotation of the mitogenome of *P. aculeata* and revealed its genomic characteristics and structural features. We analyzed relative synonymous codon usage (RSCU) and repeat sequences and discussed the possibility of chloroplast DNA transfer into the mitogenome. We also investigated RNA editing sites of protein-coding genes (PCGs) and explored their phylogenetic relationships. The results reported here provide a unique insight into the mitochondrial evolution of a cactus species, and it is also a basis for developing available genetic resources and the molecular marker-assisted breeding of *P. aculeata*.

## 2. Results

### 2.1. Mitogenome Structure of P. aculeate

When the mitogenome of *P. aculeata* was assembled, a unitig graph mediated by two repeats was generated (Figure 1A). Among them, the length of R1 is 9026 bp, and that of R2 is 7040 bp. The two repeat sequences each have four paths (p1–p4 and p5–p8, Figure 1B,E). We verified the existence of these paths in the mitogenome of *P. aculeata* using a polymerase chain reaction (PCR) experiment. In this experiment, we designed a total of six pairs of primers. The first three pairs of primers (F1 + R1, F2 + R1, F3 + R3, F3 + R2) are used to verify repeated sequence R1, and the rest (F4 + R4, F5 + R5, F6 + R4, F5 + R6) are used to verify repeated sequence R2. The distribution diagram and sequence information of these primers are shown in Figure 1C,F and Appendix A, respectively. The PCR products were consistent with the expected result (Figure 1D,G). The PCR-based experiments showed that the assembly here was correct and that were two large repeating elements in this mitogenome.

Long reads were subsequently used to further explore the authenticity of these paths (p1–p8) and their specific proportions. As shown in Appendix A, the eight paths supported by long reads are as follows: p1 (27), p2 (23), p3 (23), p4 (14), p5 (34), p6 (35), p7 (32) and p8 (18). The number in parentheses indicates the number of long reads that support this path. This further confirmed the existence of multiple mitogenome conformations. Based on this, we drew the possible isomers, as shown in Figure 2. In this study, we use the structure shown in Figure 2A for the downstream analysis, which is supported by most long reads.

Overall, *P. aculeata* mitogenome is a putative circular DNA with a length of 515,187 bp, and the GC content is 44.05% (Figure 3). Two direct repeats (DRs) with lengths of 9026 bp and 7040 bp. The accuracy of the mitogenome assembly was confirmed by mapping the ONT long reads (average 75-fold depth) onto the assembly (Appendix A).

### 2.2. Mitochondrial Gene Content

A total of 33 unique protein-coding genes (PCGs) were annotated (Table 1), including 24 core genes and nine variable genes. The core genes included five ATP synthase genes (*atp1*, *atp4*, *atp6*, *atp8*, and *atp9*), nine NADH dehydrogenase genes (*nad1*, *nad2*, *nad3*, *nad4*, *nad4L*, *nad5*, *nad6*, *nad7*, and *nad9*), four cytochrome c biogenesis genes (*ccmB*, *ccmC*, *ccmFC*, and *ccmFN*), three cytochrome c oxidase genes (*cox1*, *cox2*, and *cox3*), one membrane transport protein gene (*mttB*), one mature enzyme gene (*matR*), and one ubiquinol cytochrome c reductase gene (*cob*). The variable genes included two large subunits of ribosomal protein (*rpl5* and *rpl16*), six small subunits of ribosomal protein (*rps1*, *rps3*, *rps4*, *rps7*, *rps12*, and *rps13*), and one succinate dehydrogenase (*sdh4*). Among these, the genes *atp9* and *nad4* had two copies.

A total of 29 tRNA genes were annotated based on tRNAscan-SE. Based on sequence similarity and a previous report [32], we distinguished these tRNA genes transferred from plastid to mitochondria in the *P. aculeata* mitogenome. Firstly, a total of 13 tRNA genes are native to mitochondria, including *trnC-GCA* (mt), *trnE-UUC* (mt), *trnF-GAA* (mt), two copies of *trnG-GCC* (mt), *trnK-UUU* (mt), *trnI-CAU* (mt), *trnfM-CAU* (mt), *trnP-UGG* (mt), *trnQ-UUG* (mt), *trnS-GCU* (mt), *trnS-UGA* (mt), and *trnY-GUA* (mt). Secondly, nine tRNA genes were highly homologous to the chloroplast’s tRNA genes in sequences, including *trnD*-*GUC* (cp), *trnH*-*GUG* (cp), *trnN*-*GUU* (cp), *trnV*-*GAC* (cp), *trnW*-*CCA* (cp), two copies of *trnM*-*CAU* (cp), *trnP*-*UGG* (cp), and *trnI*-*CAU* (cp). This suggests that these sequences were transferred between organelles during evolution. Notably, we also found a bacterial-origin tRNA gene (*trnC-GCA*) with a high degree of sequence homology with previously reported genes [33,34]. Lastly, gene *trnT*-*UGU* and the remaining five copies of *trnM*-*CAU* are from unknown sources.

Based on the BLASTn program, we annotated three unique rRNA genes, *rrn5*, *rrn18*, and *rrn26*, all have two copies. The detailed location of each gene is given in Appendix A. Figure 3 shows the genome map of *P. aculeate*.

Previous studies have shown that the number of intron-contained genes varies among the mitogenomes of land plants [35]. The introns of mitochondrial genes were analyzed, and 10 genes were found to have introns (Appendix A). Among them, gene *ccmFC*, *rps3*, *trnI*-*CAU*, and *trnT-UGU* had one *cis*-intron; the genes *cox2* and *nad4* had two *cis*-introns; the gene *nad7* had four *cis*-introns; The gene *nad1* had one *cis*-intron and three *trans*-introns; the gene *nad5* had two *cis*-introns and two *trans*-introns; the gene *nad2* had three *cis*-introns and one *trans*-intron.

### 2.3. Codon Usage Analysis of PCGs

The eukaryotic genome has 64 codons that encode 20 different amino acids. All amino acids except Met and Try are encoded by multiple codons. Because of the degeneracy of codons, the usage of codons in different species varies greatly. Codon usage analysis was performed on 35 PCGs of *P. aculeata* mitogenome (Figure 4 and Appendix A). Codons with an RSCU greater than 1 are used preferentially by amino acids [36,37], and it can be found that mitochondrial PCGs have a universal codon usage preference. For example, leucine (Leu) prefers UUA codons, and its RSCU value is the highest among mitochondrial PCGs, reaching 1.64. The second was alanine (Ala), with a preference for GCU (RSCU = 1.61). In addition, UUU (Phe), AUU (Ile), and UUA (Leu) are the three most common codons in *P. aculeata*. This is possibly a kind of preference produced by *P. aculeata* after a long period of evolutionary selection.

### 2.4. Repeat Elements

The SSRs are special tandem repeats, usually no more than 6 bp [38]. The SSRs are often used for molecular marker development [39,40]. A total of 165 SSRs were identified in *P. aculeata* mitogenome (Appendix A), and 60 of them were mainly repeat units consisting of tetranucleotides, accounting for 36.36% of the total SSRs. The second was mononucleotide consisting of 44 repeat units, accounting for 26.67% of the total SSRs. In addition, 28 dinucleotide repeat units, 23 trinucleotide repeat units, eight pentanucleotide repeat units, and two hexanucleotide repeat units were detected. The number of SSRs is shown in Figure 5A.

We identified the dispersed repeats in the mitogenome of *P. aculeate*. The forward and palindromic repeats account for the vast majority. A total of 421 pairs of repetitive sequences with lengths greater than or equal to 30 were observed, including 203 pairs of forward repeat, 217 pairs of palindromic repeat, and one pair of the complementary repeat (Appendix A and Figure 5B). No reverse repeats were detected in the mitogenome of *P. aculeate*. It has been reported that these repeats may promote genome rearrangements and affect genome size [41]. In the present study, most of these repeats are less than 100 bp in length, and only 23 repeats are longer than 100 bp. Among them, the longest forward repeat was 9026 bp, followed by 7040 bp, and they were both confirmed to be associated with the alternative conformations.

The total length of these dispersed repeats is 35,740 bp, accounting for 6.94% of the whole mitogenome of *P. aculeate*, which showed an abundance of repeats. These dispersed repeats can not only cause genome recombination but also induce mutation [42].

### 2.5. Plastid DNA Insertion in Mitogenome

We observed a lot of sequences being transferred from the chloroplast genome to its mitogenome in *P. aculeata*. According to sequence similarity analysis, there were 38 homologous fragments between the mitogenome and chloroplast genome, with a total length of 26,721 bp, accounting for 5.19% of the total mitogenome (Appendix A). Among them, 10 fragments exceeded 1000 bp, we numbered them according to their length, and each started with ‘MTPT’. MTPT1 is the longest at 3962 bp. These results indicate that the chloroplast genome provides abundant non-native sequences for its mitogenome. By annotating these homologous sequences, 15 complete genes were also found on 38 homologous fragments, including six plastidial PCGs (*rpoC1*, two *rpl2*, *petN*, *psbD*, and *ndhB*) and nine tRNA genes (*trnD-GUC*, two *trnM-CAU*, *trnI*-*CAU*, *trnW-CCA*, *trnP-UGG*, *trnH-GUG*, *trnV*-*GAC*, *trnN*-*GUU*). The schematic representation of the MTPTs is shown in Figure 6. In addition, there are also some plastid gene fragments found in the mitogenome, such as *rpl23*, *rbcL*, *rpoC2, rpoB, psbC*, *ycf3*, *atpA,* and *atpF*. However, no gene has been annotated on homologous fragments such as MTPT4, MTPT8, MTPT13, and so on. Detailed information about transferred genes can be found in Appendix A.

### 2.6. RNA Editing

Based on Deepred-mt, the RNA editing events of 33 PCGs from mitogenome were identified. The 25 species on the page were selected as the database. Using this criterion, 362 potential RNA editing sites were identified (Figure 7 and Appendix A). There are great differences in the number of sites edited by different genes. Among mitochondrial genes, the *ccmB* gene identified 30 RNA editing sites, the most among all genes. This was followed by *ccmC*, with 28. However, the *rps13* and *rps7* genes only identified one RNA editing site.

Studies have shown that a large number of C to U substitutions occur during mitochondrial DNA transcription in higher plants [43]. By examining the types of RNA editing events, it was found that all the editing sites in the mitogenome of *P. aculeata* were C-U editing. This is consistent with previous statements. Notably, we found that the stop codons of two genes (*atp6* and *atp9*) and the start codons of four genes (*cox2*, *nad1*, *nad4L*, and *nad7*) were created through RNA editing events and were supported by the high confidence of Deepred-mt. The new start and stop codons are usually generated to encode proteins that are more conserved and homologous to the corresponding proteins of other species, thus allowing better expression of the genes in the mitochondria [44].

According to the result of Deepred-mt, the C to U RNA editing events generated three stop codons in two genes, namely *atp6* and *atp9*. The *atp6* is CAA to UAA, and the *atp9* is CGA to UGA. The start codons of *cox2*, *nad1*, *nad7*, and *nad4L* are created by RNA editing, which was achieved by editing ACG to AUG. RNA editing in plants may lead to deleterious phenotypes and even death [45]. Therefore, it is very important to study RNA editing in plants.

### 2.7. Phylogenetic Inference

To investigate the phylogenetic position of *P. aculeata*, we downloaded mitogenome sequences of 24 species from GenBank, including 21 closely related species and two outgroups. The phylogenetic tree (Figure 8) showed that Caryophyllaceae, Amaranthaceae, and Chenopodiaceae were clustered into one clade, and Aizoaceae, Nyctaginaceae, and Cactaceae were clustered into another clade. Polygonaceae and Nepenthaceae each form a separate clade. The phylogenetic tree is consistent with the Angiosperm Phylogeny Group IV (APG IV) classification system; that is, *P. aculeata* belongs to the Cactaceae family of Caryophyllales, which is closely related to Aizoaceae and Nyctaginaceae. Most of the nodes of the phylogenetic tree in this study had relatively high bootstrap values, indicating that the phylogenetic analysis had high reliability.

### 2.8. Synteny Analysis

Synteny analysis deals with the arrangement of homologous genes or sequences. Based on Mauve (version 20150226) software [46], we performed synteny analysis on the mitogenomes of *P. aculeata* and four related species (*Tetragonia tetragonoides*, *Mirabilis jalapa, Sesuvium portulacastrum*, and *Bougainvillea spectabilis*) (Appendix A). These species exhibit many common colinear mitogenomic sequences. However, these collinear sequences have an irregular arrangement, showing a rearrangement of the mitogenome. These results also indicate that the conformation of mitogenome is extremely non-conserved. As shown in Figure 9, there are only 86 shared collinear blocks among these five mitogenomes, with lengths ranging from 115 bp to 3780 bp. In contrast, a large number of regions (mainly non-coding regions) show high species-specific levels.

## 3. Discussion

Our study involves the analysis of the *P. aculeata* mitogenome, which is 515,187 bp long (represented in Figure 3). It is important to note that there is a possibility of alternative configurations and a multi-chromosomal architecture. Although Unicycler only obtained one independent and circular contig by resolving repeats with long reads, we showed that there are potential multiple conformations based on PCR and long reads. The 9026 bp and 7040 bp repetitive elements could possibly be the key to recombination. Furthermore, we found that the similarities between the two repetitive units of the above dispersed repeats are 100% (Appendix A). A high similarity could be indicative of ongoing gene conversion and concerted evolution [47]. The plant mitogenome undergoes recombination with the aid of large repetitive segments present in multiple locations within mitochondrial DNA. These large repetitive elements were found in many plant mitogenomes. For example, the mitogenome of *Thuja sutchuenensis* consists of one circular contig and three linear contigs (1,390,975 bp, 519,836 bp, and 293,570 bp) with overlapping regions, which was confirmed though PCR [48]. A 21,809 bp repetitive element was also found in the *Taraxacum mongolicum* mitogenome, which was also confirmed through PCR [49]. A total of five pairs of repetitive elements have been shown to mediate recombination in dandelion mitogenomes [49]. Short repetitive elements also confirmed such genome recombination. According to research, *Ipomoea batatas* [50], *Nymphaea colorata* [51], *Salvia miltiorrhiza* [52], and *Ginkgo biloba* [53] can all produce different alternative configurations through short repeats. It was shown that repetitive sequences mediate the reorganization of the mitochondrial genome with different frequencies [51]. Similar phenomena were observed in *Silene latifolia* [54] and *Mimulus guttatus* [55]. Here, we confirmed that two pairs of long repeats were involved in genome recombination based on long reads and PCR experiments in the *P. aculeata* mitogenome. This recombination involving long repeats may not have caused functional damage to the mitogenome of *P. aculeata*, and this recombination is likely to be reversible. However, it is not known whether some short-repeats-mediated recombination exists, and in the future, high sequencing depth data will be required to achieve low-frequency recombination detection.

A total of 165 SSRs were identified in the mitochondrial genome of *P. aculeata*. The number of SSRs in the form of a single-nucleotide (A/T) polymer was the largest, accounting for 26.06% of all SSRs. The same was true in the mitogenome of *Phaseolus vulgaris* [16]. This may be related to the low GC content in mitochondria. Secondly, we found that the mitogenome contained a large number of dispersed repeats. They are mainly forward repeats and palindromic repeats. The ratio between them is close to 1:1. A similar pattern was found in *Gleditsia sinensi*, which had 26 forward repeats and 24 palindromic repeats [36].

The plant mitogenome is more prone to accept and integrate foreign DNA because of the special properties of its genome structure and evolutionary process [31]. Foreign DNA is widely found in the plant mitogenome [56]. Some studies have shown that the division and partial gene transfer of the ribosomal protein gene *rpl2* has been found in the plant mitogenome, which provides direct evidence for the easy acceptance and integration of foreign DNA in the plant mitotic genome [57]. For angiosperms, there is extensive communication of genetic material between the two organelles in the cell, mainly the migration of DNA from the chloroplast to the mitogenome [17]. Mitochondrial DNA transfer into the chloroplast genome has been found only in *Anacardium occidentale* [58]. In *P. aculeata*, we found 38 MTPTs. Among them, the MTPT1 is the longest at 3962 bp. Research on MTPTs has been widely reported. In *Suaeda glauca*, 26.87 kb of MTPTs accounted for 5.18% of its mitogenome [17]. These large MTPTs were thought to influence the evolution of eukaryotes extensively and promote genetic diversity. Additionally, one study showed that the origin of tRNAs in plant mitochondria is divided into two parts: one is inherited from the mitochondrial ancestors, and the other is acquired from chloroplasts by HGT [59].

Based on sequence similarity and a previous report [32], we can trace the tRNA genes transferred from plastid to mitochondria in the *P. aculeata* mitogenome. In *P. aculeatatrn*, *trnD*-*GUC*, *trnH*-*GUG*, *trnN*-*GUU*, *trnV*-*GAC*, *trnW*-*CCA*, *trnM*-*CAU*, *trnP*-*UGG*, and *trnI*-*CAU* were transferred from chloroplast genes. It makes up 42.11 percent of all tRNAs. Of these tRNA genes, *trnW-CCA* (cp) is broadly present in the mitogenomes of other angiosperms, and they appear to be homologous to chloroplasts. However, *Amborella trichopoda* has mitochondrial native *trnW-CCA*, and this is not seen in other angiosperms [34]. Additionally, *trnP-UGG* (cp) may also be functional, as reported by Richardson et al. [32]. Furthermore, we also found a bacterial-origin tRNA gene (*trnC-GCA*) in the mitogenome of *P. aculeata*. The bacteria-derived *trnC-GCA* has been reported previously [33,34], but it is not very commonly found in plant mitogenomes.

In *P. aculeata*, we found that MTPTs contained 15 complete genes, including six PCGs and nine tRNA genes. In addition, some chloroplast PCGs lost their integrity during evolution and migrated from the chloroplast genome to the mitogenome in the form of fragments. There is a similar situation in the mitogenome of *Suaeda glauca* [17]: chloroplast genome PCGs may migrate among organelles in the form of partial sequences. Therefore, it was found that tRNA genes are more conserved than PCGs and play an important role in the evolution of plants. In general, the chloroplast genome of *P. aculeata* provides abundant foreign sequences for its mitogenome, which increases the diversity of mitochondrial DNA sequence sources.

In eukaryotes, the coding regions for transcribed RNA undergo a process called RNA editing by adding, missing, or replacing bases [60]. Studies have found that RNA editing is widespread in higher plant mitochondria, and it is one of the essential steps for gene expression in the plant mitogenome [61,62]. For most angiosperms, the chemistry of RNA editing is a deamination reaction in which a site-specific cytosine (C) becomes uracil (U) [63]. Studies have shown that RNA editing of mitochondrial genes may regulate plant cytoplasmic inheritance-related traits [64]. In *P. aculeata*, we recorded a total of 362 RNA editing events, and most of the RNA editing sites occurred at the first position or second position, which is similar to the situation in most plants [16,36,65,66]. By identifying the RNA editing sites, the gene function of the new code can be effectively predicted. Notably, we found that the stop codons of two genes (*atp6* and *atp9*) and the start codons of four genes (*cox2*, *nad1*, *nad7*, and *nad4L*) were created by RNA editing events and were supported by the high confidence of Deepred-mt. These new start and stop codons encode more conserved and homologous proteins, promoting better gene expression in mitochondria [44].

## 4. Materials and Methods

### 4.1. Plant Sampling, DNA Extraction, and Sequencing

We collected the fresh leaves of *P. aculeata* from Nanshan Botanical Garden, Chongqing (Geospatial coordinates: N29.561209, E106.635764). The samples were identified by Professor Jie Yu based on morphological characteristics, and they have been deposited in the Herbarium of Southwest University in Chongqing, with the accession number SWU-YJ-009. The total genomic DNA was extracted using the CTAB method [67], and the total DNA was fragmented using ultrasound. The DNA library with an insertion size of 350 bp was constructed with the NEBNext^®^ library construction kit (NEB, Ipswich, MA, USA) [68] and was sequenced by using the HiSeq Xten PE150 sequencing platform (Novogene, Beijing, China). Sequencing produced a total of 5.35 Gb of raw data, including 17,835,750 raw reads. Clean data were obtained using Trimmomatic (v0.36) [69]: low-quality sequences with more than 5% base N were eliminated, and more than 5% base quality values Q < 19. Finally, we obtained a total of 17,631,944 clean reads. The same DNA sample was also used for Oxford Nanopore sequencing based on the promethION platform. A total of 13.4 Gb of sequence reads were obtained. The mean read length of sequencing was 17.52 kb, and the N50 was 26.50 kb.

### 4.2. Genome Assembly

To begin, we utilized Flye software (v2.9.1-b1780) [70] to conduct de novo assembly of the Oxford Nanopore long reads derived from *P. aculeata*. The parameters used for this assembly were ‘--min-overlap 8000’. We then employed the BLASTn (v2.2.30+) [71] program to identify the draft mitogenome of the *P. aculeata* based on the assembled contigs. To achieve this, we constructed a database for the assembled sequences using makeblastdb and selected the conserved mitochondrial genes from *Liriodendron tulipifera* (NC_021152.1) as a query sequence to identify contigs that contain conserved mitochondrial genes. A total of six unitigs were identified. These six unitigs are interconnected to form a branching genome structure. Finally, we mapped both the short reads and long reads onto these contigs and retained all the mapped reads using BWA (v0.7.12-r1039) [72] and SAMTools software (v0.1.19) [73].

Due to the possibility of chloroplast-homologous regions of the mitogenome being replaced by chloroplast counterparts during polishing, we used a combination of Illumina short-reads and Nanopore long reads to perform hybrid assembly using Unicycler (v0.4.8) [74], with the parameters of ‘- kmers 27, 53, 67, 71, 89, 99, 105’. Initially, the mapped Illumina short-reads were assembled using spades [75]. Then, repetitive regions of the assembly were resolved using minimap2 [76] with the Nanopore long reads. The GFA format files produced by Unicycler were visualized using Bandage (v0.8.1) [77]. Finally, Unicycler generated one circular contig, and as this contig was assembled using short reads, no additional polish was necessary.

### 4.3. Validating of Genome Structure

We use two strategies to determine the potential structure of the mitogenome. First, we use long reads to detect whether the two long repetitive elements mediate the genome’s recombination. Specifically, we extracted the sequences of these two pairs of repetitive elements and their flanking regions of 500 bp from the genome assembled by Unicycler, which represents the reference sequence supporting this master circle structure. Subsequently, we simulate the sequence of possible recombination by swapping the flanking regions. Next, we used makeblastdb to construct a database of long reads, and the major and recombinant sequences were together used as query sequences using the BLASTn program with the parameter ‘-evalue 1e-10 -outfmt 6 -num_threads 10 -max_hsps 1’. Finally, we counted those long reads that completely spanned these query sequences and used them as reads that support this structure.

The second strategy is based on PCR experiments. We designed specific primers at several joints to verify the authenticity of this bifurcated structure. The primers were designed using the Primer designing tool on NCBI (https://www.ncbi.nlm.nih.gov/tools/primer-blast/; accessed on 12 January 2023), and the parameters are default. We extracted the total DNA, and the amplifications were carried out in a Pro-Flex PCR system (Applied Biosystems, Waltham, MA, USA). The final volume of PCR amplification was 25 µL, including 2 µL template DNA, 1 µL forward primer, 12.5 µL 2 × Taq PCR Master Mix, and 9.5 µL ddH_2_O. The following amplification conditions: denaturation at 94 °C for 5 min, 30 cycles for 30 s at 94 °C, 30 s at 58 °C, 60 s at 72 °C and 72 °C for 5 min as the final extension. The PCR amplicons were visualized using 1% agarose gel electrophoresis, and the single bright bands were cut and sent to the Sangon Biotech (Shanghai, China) Co., Ltd. for Sanger sequencing.

### 4.4. Genome Annotation

The chloroplast genome of *P. aculeata* [78] was published before with the accession number MW553064 (https://www.ncbi.nlm.nih.gov/nuccore/MW553064; accessed on 12 January 2023). We used CPGAVAS2 [79] with the customized option; the *P. aculeata* chloroplast genome (NC_062888.1) was used as the reference genome. The mitogenomes of *Liriodendron tulipifera* (NC_021152.1) and *Beta macrocarpa* (NC_015994.1) were used as the reference genome, and Geseq (https://chlorobox.mpimp-golm.mpg.de/geseq.html; accessed on 12 January 2023) [80] was used to annotate the mitogenome of *P. aculeata*. Apollo software (v1.11.8) [81] was used to manually edit and correct problematic annotations. In addition, all tRNA genes in the mitogenome were annotated using tRNAscan-SE software (v2.0) [82], and all rRNA were annotated using BLASTn software [71].

### 4.5. Codon Usage of Mitochondrial Genes

PhyloSuite software (v1.2.2) [83] was used to parse the GenBank format file of the *P. aculeata* mitogenome, and we extracted the protein-coding gene (PCGs). Mega 7.0 software [84] was used to analyze the codon usage of mitochondrial PCGs by calculating RSCU values.

### 4.6. Analysis of Repeat Elements

The simple sequence repeats (SSRs) were obtained via the online website MISA (https://webblast.ipk-gatersleben.de/misa/; accessed on 12 January 2023) [85]; the parameters of the minimum numbers of mono-, di-, tri-, tetra-, penta-, and hexanucleotides were set as 10, 5, 4, 3, 3, and 3, respectively. In addition, we used REPuter (https://bibiserv.cebitec.uni-bielefeld.de/reputer/; accessed on 12 January 2023) [86] to calculate forward, reverse, palindromic, and complementary repeat sequences, with the following settings: hamming distance of three and minimal repeat size of 30 bp, and e-value was limited to less than 1 × 10^−5^. The visualization of the repeat elements was accomplished by using the Circos [87] package.

### 4.7. Identification of Homologous Sequences among Organelle Genomes

The BLASTn program was performed on the two organelle genomes of *P. aculeata*, and the mitogenome was used for building databased with makeblastdb, and the chloroplast genome was used as query sequence with the following parameters: ‘-evalue 1e-10-word_size 7-outfmt 6’. Finally, the BLASTn results were visualized using the ‘Advanced Circos’ module in TBtools (v1.18) [88].

### 4.8. The Prediction of RNA Editing Sites

We used Deepred-mt [89], a tool for predicting C to U RNA editing sites based on the convolutional neural network (CNN) model, to predict C to U RNA editing sites. All mitochondrial protein-coding genes were extracted from the mitogenome and used for prediction, and we selected results with probability values greater than 0.9.

### 4.9. Phylogenetic Inference

According to the genetic relationship, the related species of *P. aculeata* were selected, and their complete mitogenome sequences were downloaded. The species information is listed in Appendix A. We downloaded the mitogenomes of 24 closely related species and two outgroups (*Tolypanthus maclurei* and *Malania oleifera*) on NCBI (https://www.ncbi.nlm.nih.gov/; accessed on 12 January 2023) nucleotide database. A total of 31 orthologous PCGs among the analyzed species were identified and extracted by using PhyloSuite (v1.2.2) [83]. The corresponding nucleotide sequences were aligned using MAFFT (v7.471) [90]. Next, these aligned sequences were concatenated and used to construct the phylogenetic trees. The maximum likelihood (ML) method was implemented in RAxML (v8.2.4). The parameters were “raxmlHPC-PTHREADS-SSE3 -f a -N 1000 m GTRGAMMA -x 551314260 -p 551314260”. The bootstrap analysis was performed with 1000 replicates. Finally, the result was uploaded to the ITOL software (v4) [91] for beautification.

### 4.10. Synteny Analysis

We chose four closed species: *Bougainvillea spectabilis* (NC_056281.1), *Tetragonia tetragonoides* (MW971440.1), *Sesuvium portulacastrum* (MK922288.1), and *Mirabilis jalapa* (NC_056991.1) for the colinear analysis with *P. aculeata.* Based on the progressive Mauve module embedded in Mauve (v.20150226) software [46], the above five mitogenomes were analyzed with default parameters. For the calculated conserved collinear blocks, we only reserved the blocks with lengths greater than 100 bp.

## 5. Conclusions

In this study, we report the complete mitogenome of *P. aculeata*. The genome length is 515,187 bp and its GC content is 44.05%. Long reads and PCR confirmed that there are two repetitive sequences in the *P. aculeata* mitogenome that promote the rearrangement of the mitogenomes and generate multiple isomers. The mitogenome of *P. aculeata* was studied in detail from many aspects, which provided important genetic resources for the phylogenetic study of *P. aculeata* and laid a foundation for the mitogenome study of the Cactaceae family.

## Figures and Tables

**Figure 1 ijms-24-08366-f001:**
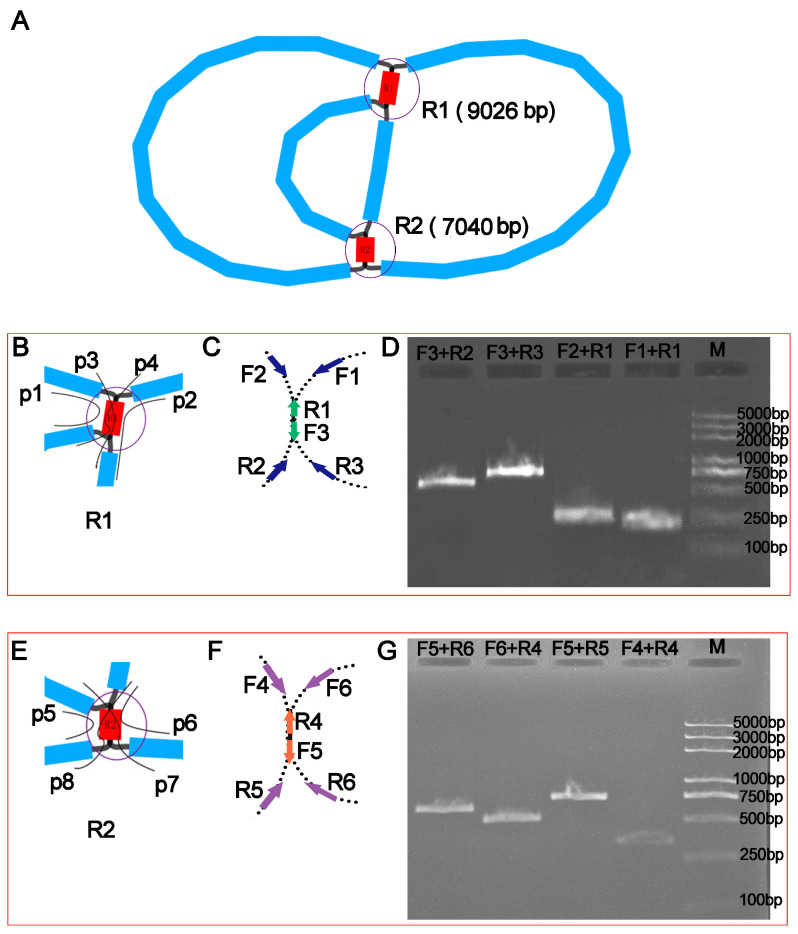
Assembly graph of *P. aculeata* mitogenome and schematic diagram of the designed specific primer pairs. (**A**) Assembly graph visualized using Bandage. The assembly contains six contigs that include two DRs (R1 and R2, highlighted in red). R1 has four paths (p1–p4), as shown in panel (**B**). Panel (**C**) shows the design of three pairs of primers for the PCR experiments. These three pairs are used to verify the repeat sequence R1 (9026 bp). Among them, F1, F2, R2 and R3 are represented by the blue arrow. On the contrary, R1 and F3 are represented by the green arrow. Panel (**D**) shows the results of agarose gel electrophoresis, as expected. R2 has four paths (p5–p8), as shown in panel (**E**). Panel (**F**) shows the design of three pairs of primers for the PCR experiments. These three pairs are used to verify the repeat sequence R2 (7040 bp). Among them, F4, F6, R5 and R6 are represented by the blue arrow. On the contrary, R4 and F5 are represented by the green arrow. Panel (**G**) are the results of agarose gel electrophoresis, also as expected.

**Figure 2 ijms-24-08366-f002:**
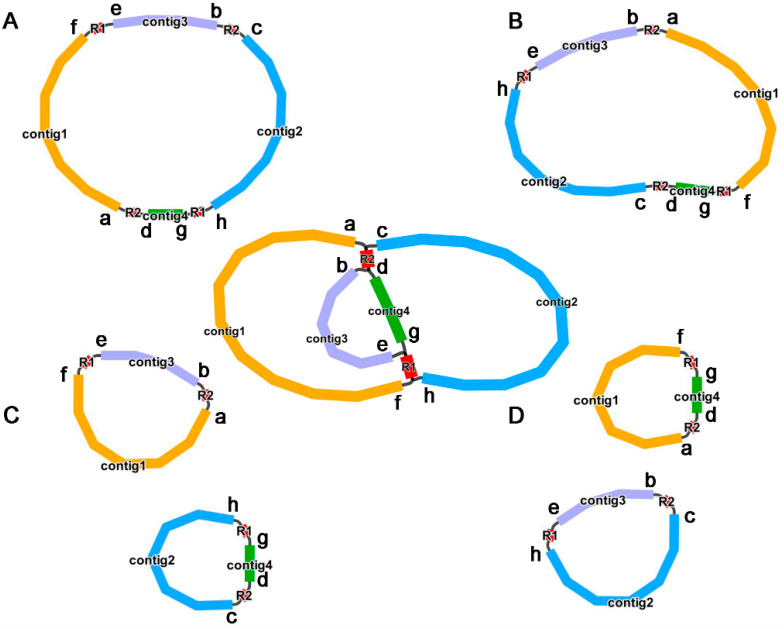
Potential isomers of *P. aculeata* mitogenome inferred from PCR experiments and long reads. The initial assembly is shown in the middle of the figure, with (**A**–**D**) representing the four possible isomers formed after solving the paths of the two pairs of repeating regions (R1 and R2). We use lowercase letters (**a**–**h**) to mark breakpoints at several nodes to describe several paths, where (**e**,**f**) stands for p1, (**g**,**h**) stands for p2, (**e**–**h**) stands for p3, (**f**,**g**) stands for p4, (**a**,**b**) stands for p5, (**c**,**d**) stands for p6, (**a**–**d**) stands for p7 and (**b**,**c**) stands for p8. These isomers contain two ‘master circle’ structures (**A**,**B**), which differ in sequence order, and two structures of two separate small circles (**C**,**D**). In this study, we use the structure shown in panel (**A**) for the downstream analysis, which is supported by most long reads.

**Figure 3 ijms-24-08366-f003:**
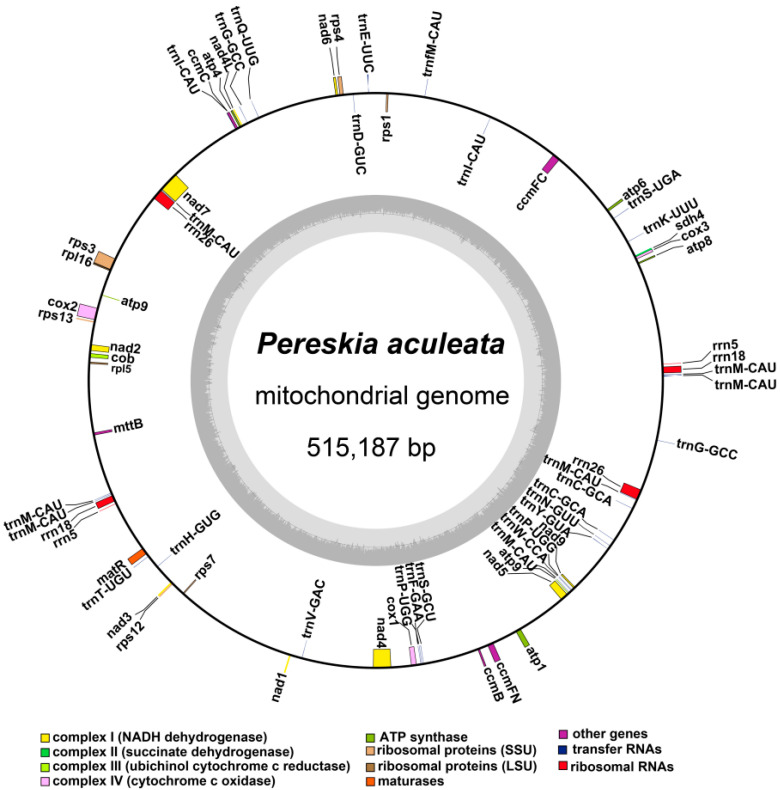
The putative circular mitogenome maps of *P. aculeata*. The genomic features inside and outside the circle represent the clockwise and counterclockwise chains on the transcription, respectively. Different color blocks represent different functional gene groups.

**Figure 4 ijms-24-08366-f004:**
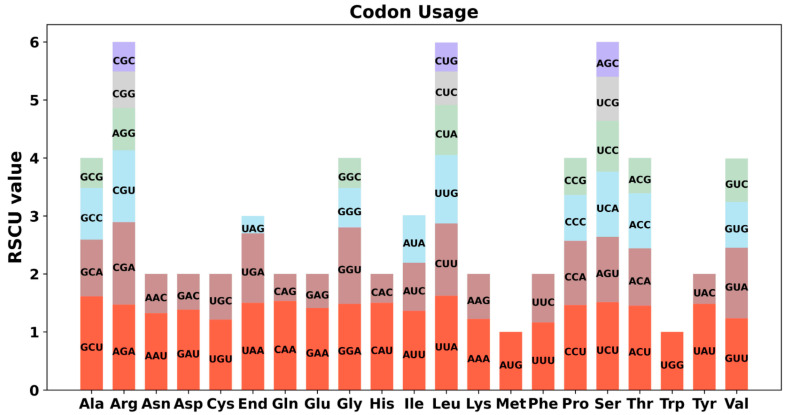
Codon usage bias of mitochondrial PCGs of *P. aculeata.* The RSCU refers to relative synonymous codon usage.

**Figure 5 ijms-24-08366-f005:**
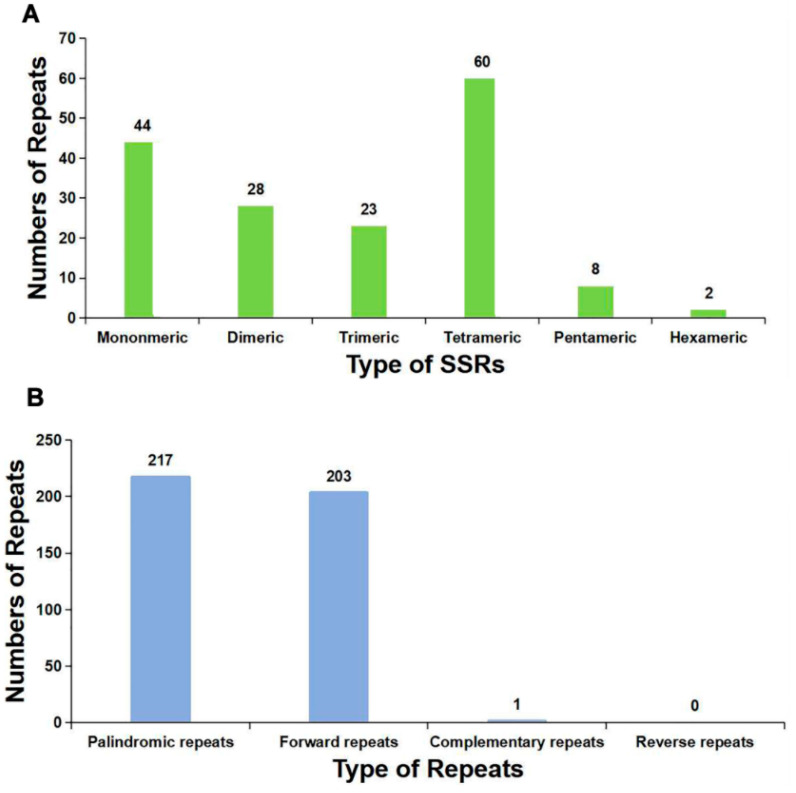
The SSRs and dispersed repeats identified in the mitogenomes of *P. aculeata.* (**A**) The SSRs identified in the *P. aculeata* mitogenomes. Each column represents different nucleotide repeated units displayed in different colors. (**B**) Dispersed repeats (≥30 bp) identified in the *P. aculeata* mitogenomes.

**Figure 6 ijms-24-08366-f006:**
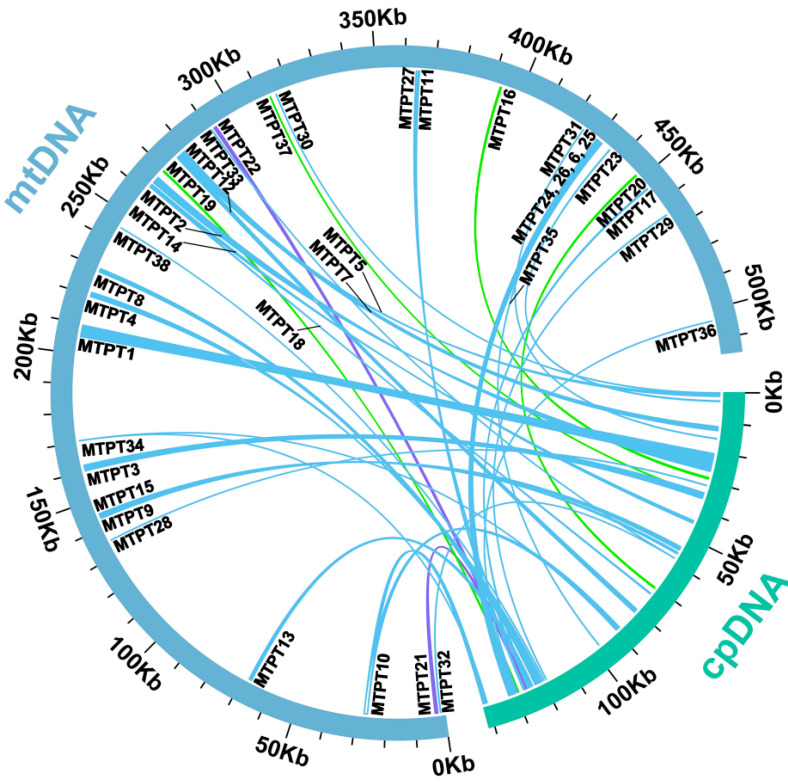
Schematic representation of homologous sequences between chloroplast genome and mitogenomes in *P. aculeata.* The blue arcs represent mitogenomes, the green arcs represent chloroplast genomes, and the lines between arcs correspond to homologous genome segments. A total of 38 MTPTs were identified, and their numbers are given in the figure. Among them, lines with a percentage of sequence similarity above 90 are blue, those between 80 and 90 are green, and those less than 80 are purple. A total of 17 complete genes migrated to the mitogenome, including 8 PCGs and 9 tRNA genes. These complete genes were labeled next to the corresponding MTPT. Some homologous sequences are in the IR region of the plastome. We count them according to unique locations on the mitogenome.

**Figure 7 ijms-24-08366-f007:**
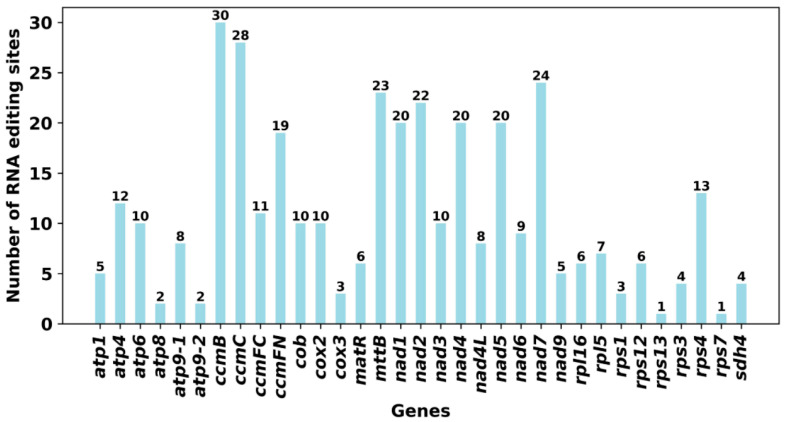
Characteristics of the RNA editing sites identified in PCGs of *P. aculeata* mitogenome. Number of RNA editing sites predicted by individual PCGs using Deepred-mt. The abscissa shows the name of the gene, and the ordinate shows the number of edited sites.

**Figure 8 ijms-24-08366-f008:**
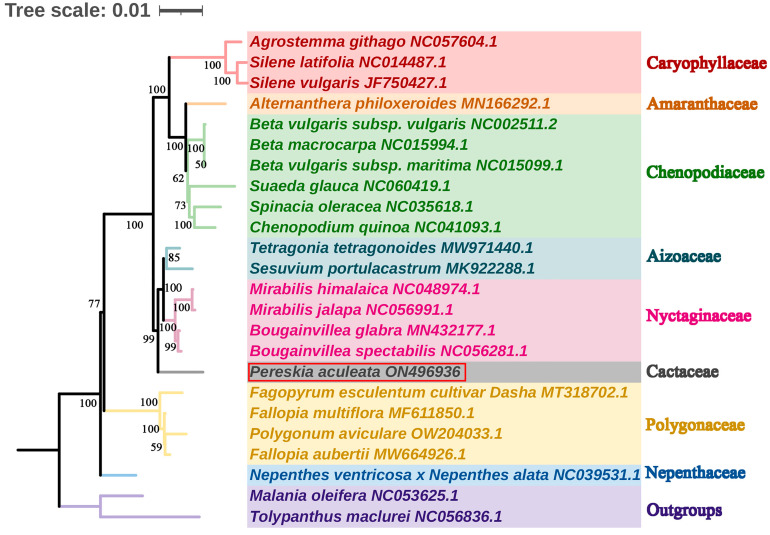
Phylogenetic tree constructed from the shared mitochondrial PCGs of 24 species. The *Tolypanthus maclurei*, and *Malania oleifera* were used as the outgroups. The number at each node showing the bootstrap support value.

**Figure 9 ijms-24-08366-f009:**
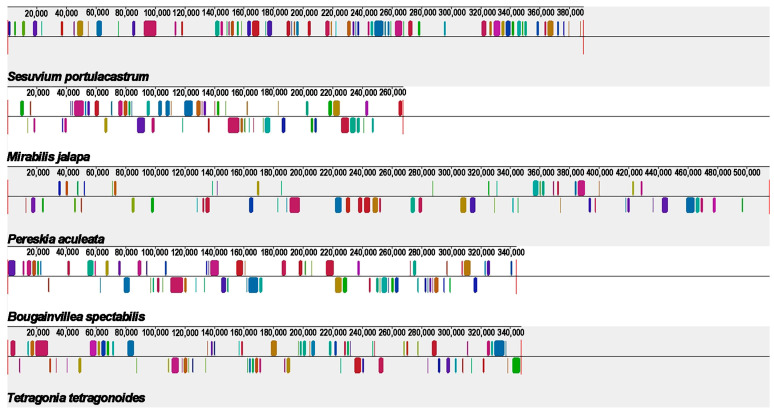
Synteny analysis of five mitogenomes. Only collinear blocks over 0.1 kb in length are retained.

**Table 1 ijms-24-08366-t001:** Gene composition in the mitogenome of *P. aculeate*.

Group of Genes	Name of Genes
ATP synthase	*atp1*, *atp4*, *atp6*, *atp8*, *atp9* (2)
NADH dehydrogenase	*nad1*, *nad2*, *nad3*, *nad4* (2), *nad4L*, *nad5*, *nad6*, *nad7*, *nad9*
Ubichinol cytochrome c reductase	*cob*
Cytochrome c biogenesis	*ccmB*, *ccmC*, *ccmFC*, *ccmFN*
Cytochrome c oxidase	*cox1*, *cox2*, *cox3*
Maturases	*matR*
Transport membrane protein	*mttB*
Large subunit of ribosome	*rpl5*, *rpl16*
Small subunit of ribosome	*rps1*, *rps3*, *rps4*, *rps12*, *rps13*, *rps7*
Succinate dehydrogenase	*sdh4*
Ribosome RNAs	*rrn18* (2), *rrn26* (2), *rrn5* (2)
Transfer RNAs	*trnC-GCA* (2), *trnD-GUC*, *trnE-UUC*, *trnF-GAA*, *trnM-CAU* (7), *trnH-GUG*, *trnI*-*CAU*(2), *trnK-UUU*, *trnN-GUU*, *trnP-UGG* (2), *trnQ-UUG*, *trnS-GCU*, *trnS-UGA*, *trnV-GAC*, *trnW-CCA*, *trnY-GUA*, *trnG-GCC* (2), *trnT-UGU*, *trnfM-CAU*

Note. The numbers in parentheses represent copy numbers of genes.

## Data Availability

The mitogenome sequence is available in the nucleotide database of NCBI (https://www.ncbi.nlm.nih.gov/; accessed on 12 January 2023) with accession numbers: NC_067638.1 or ON496936.1. The sequencing reads used for mitogenome assembly in this study have been released on the NCBI with those accession numbers: PRJNA715621 (BioProject), SAMN18357766 (BioSample), and SRR21847255 (SRA). The sample has been deposited in the Herbarium of Southwest University in Chongqing, with the accession number SWU-YJ-009. The mitochondrial long reads can be found in Appendix A.

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
