# Peer review of "Assembly of the Complete Mitochondrial Genome of Pereskia aculeata Revealed That Two Pairs of Repetitive Elements Mediated the Recombination of the Genome"

_ijms, 2023, doi:10.3390/ijms24098366_

Round 1
Reviewer 1 Report
Abstract
Lines 13-14) “Pereskia aculeata is a plant-based food resource that has both medicinal and edible values, with antinociceptive activity.” This is not a resource but a species with potential economic value. Reviewer suggests opening the Abstract as follows: “Pereskia aculeata is a potential new crop species that has both food and medicinal (antinociceptive activity) properties.”
Lines 15-17) “…used as the experimental material to address the gap by sequencing, assembling, and annotating its mitochondrial genome (mitogenome) to understand its genetic information and developmental relationships.” Suggest change to “…studied to sequence the mitochondrial genome (mitogenome) and to ascertain the assembly, informational content ,and developmental expression of the mitogenome.”
Line 18) “…is a circular genome measuring…” change to “is circular and measures…”.
Lines 23-24) “…mediated mitogenome recombination with high frequency.” Change to “…were highly involved with the mitogenome recombination process.”.
Line 24) “…two organelle…” change to “…mitogenome and chloroplast genome…”.
Line 26) “…Cactaceae, the decoding…” a new sentence is needed here “…Cactaceae. The decoding…”.
Keywords: Should be alphabetized.
Introduction
Lines 36 and 39) The use of contrasting prepositions “On the one hand….” and “On the other hand…” implies that the described features are in contrast to each other. Reviewer suggests the sentences be rewritten to simply describe the defining nutritional features of P. aculeata.
Lines 47-48) “Mitochondria is a semi-autonomous organelle with its genetic system, which is the main place for energy conversion in cells.” Change to “The mitochondrion is a semi-autonomous eukaryotic organelle that has a small genome (mitogenome) that interacts with the nuclear and chloroplast genome to provide the biochemical machinery for energy conversion.”
Line 50) “In addition, mitochondria are also involved…” redundant; change to “Mitochondria are also involved…”.
Line 52) “…closely related to cytoplasmic male sterility (CMS)…” change to “…involved in maternally-inherited male sterility (CMS)…”
Line 52) “Modern genetic breeders…” change to “modern plant breeders…”.
Line 54) “…more complex and difficult to sequence…”. More than the nuclear genome? A sentence to describe how the mitogenome is more complex and difficult to sequence than the nuclear genome would be appropriate here.
Lines 55-57) How are chloroplast and plastid genomes distinct?
Lines 59-60) “From these different conformations, the mitogenome structure in higher plants is unstable.” Change to “The mitogenome structure in higher plants is also unstable.”.
Line 61) Remove “unfortunately”.
Line 68) “…of cacti species,…” change to “…of a cactus species,…”.
Results
Lines 74 and 75) The mitogenome lengths in bp have a space in them that confuses the information. For example, 9, 026 bp, and 7, 040 bp should be written 9,026 bp and 7,040 bp with no internal spaces in the numbers. This same issue (spaces within numbers) occurs elsewhere in the manuscript, including figures, and should also be revised accordingly.
Line 77) “…Chain Reaction (PCR) experiment.” Change to “…chain reaction (PCR).”.
Line 81) “…Table S2 respectively…” change to “…Table S2, respectively.”.
Line 81) “The PCR products are…” change to “The PCR products were…”.
Line 83) “…here…” delete.
Figure 1 legend) Should be written in the present tense since it refers to the representation of results.
Line 100) “Next, we used long-reads…” change to “Long reads were subsequently used…”.
Lines 120-121) “…are potentially to mediate mitogenome recombination and to form putative multiple conformations.” This statement should be moved to the Discussion section since it is conjecture.
Line 129) “…plant and 9 variable genes…” this is confusing and should be explained further.
Table 1) The parentheses indicate the numbers of identified genes in the mitogenome. The reviewer suggests that the “x2” notation be shortened to simply “2”.
Line 150) “…are unknown sources…” change to “…are from unknown sources…”.
Line 152) “rrn26, and they all have two copies…” change to “…rrn26 all have two copies…”.
Line 158) “The gene…” has a This may be relatedsequence homology between the P. aculeata mitogenome and chloroplast genome, suggesting that these sequences were transferred between organelles during evolution.”.
Line 210) “The MTPT1 is the longest at 3, 962 bp.” Change to “MTPT1 is the longest at 3,962 bp.”.
Line 211) “..foreign…” suggest change to “…non-native…”.
Line 217) “…and so on…” delete.
Line 236) “…1…” change to “…one…”.
Line 276) “…its…” delete.
Lines 277-278) “It turns out that there are a lot of collinear blocks between them.” Change to “These species exhibit many common colinear mitogenomic sequences.”.
Line 278) “This may be related…” change to “Sequence colinearlity may be due…”.
Line 280) “…un-conserved…” reviewer believes that “…nonconserved…” is more appropriate.
Lines 281-284) The statements here are confusing to the reviewer and should be rewritten to be clearer.
Discussion
Line 291) “…It's…” change to “…it is…”.
Lines 300 and 302) “ … experiment…” delete.
Lines 302-303) “A total of five pairs of repetitive elements have been shown to mediate recombination in dandelion mitogenomes.” A citation is needed here to verify the presented information.
Lines 303-304) “Besides, the short repetitive elements are also confirmed to mediate genome recombination.” Change to “Short repetitive elements also confirmed such genome recombination.”.
Line 329) “Based…” suggest a new paragraph starting with this sentence.
Line 340) “In most plant lineages, the tRNA gene is lost after it is transferred from bacteria.” This statement needs a citation.
Materials and Methods
Line 404) “…we extract...” change to “…we extracted…”.
Line 411) “…we count those long-reads that completely span these query sequences,,,” change to “…we counted those long-reads that completely spanned these query sequences…”.
Line 418) “…is 25 µL…” change to “…was 25 µL…”.
Line 445) “…e-value is limited to less than 1e-05…” change to “…e-value was limited to less than 1e-05…”.
Line 463) “And then, a total of 31…” change to “A total of 31…”.
Conclusions
Lines 482-484) “There are two repetitive sequences in the P. aculeata mitogenome, which promote the rearrangement of the mitogenomes and generate multiple conformations, which was confirmed by long-reads and PCR experiments.” Change to “Long reads and PCR confirmed that there are two repetitive sequences in the P. aculeata mitogenome that promote the rearrangement of the mitogenomes and generate multiple motifs.”
Abstract
Lines 13-14) “Pereskia aculeata is a plant-based food resource that has both medicinal and edible values, with antinociceptive activity.” This is not a resource but a species with potential economic value. Reviewer suggests opening the Abstract as follows: “Pereskia aculeata is a potential new crop species that has both food and medicinal (antinociceptive activity) properties.”
Lines 15-17) “…used as the experimental material to address the gap by sequencing, assembling, and annotating its mitochondrial genome (mitogenome) to understand its genetic information and developmental relationships.” Suggest change to “…studied to sequence the mitochondrial genome (mitogenome) and to ascertain the assembly, informational content ,and developmental expression of the mitogenome.”
Line 18) “…is a circular genome measuring…” change to “is circular and measures…”.
Lines 23-24) “…mediated mitogenome recombination with high frequency.” Change to “…were highly involved with the mitogenome recombination process.”.
Line 24) “…two organelle…” change to “…mitogenome and chloroplast genome…”.
Line 26) “…Cactaceae, the decoding…” a new sentence is needed here “…Cactaceae. The decoding…”.
Keywords: Should be alphabetized.
Introduction
Lines 36 and 39) The use of contrasting prepositions “On the one hand….” and “On the other hand…” implies that the described features are in contrast to each other. Reviewer suggests the sentences be rewritten to simply describe the defining nutritional features of P. aculeata.
Lines 47-48) “Mitochondria is a semi-autonomous organelle with its genetic system, which is the main place for energy conversion in cells.” Change to “The mitochondrion is a semi-autonomous eukaryotic organelle that has a small genome (mitogenome) that interacts with the nuclear and chloroplast genome to provide the biochemical machinery for energy conversion.”
Line 50) “In addition, mitochondria are also involved…” redundant; change to “Mitochondria are also involved…”.
Line 52) “…closely related to cytoplasmic male sterility (CMS)…” change to “…involved in maternally-inherited male sterility (CMS)…”
Line 52) “Modern genetic breeders…” change to “modern plant breeders…”.
Line 54) “…more complex and difficult to sequence…”. More than the nuclear genome? A sentence to describe how the mitogenome is more complex and difficult to sequence than the nuclear genome would be appropriate here.
Lines 55-57) How are chloroplast and plastid genomes distinct?
Lines 59-60) “From these different conformations, the mitogenome structure in higher plants is unstable.” Change to “The mitogenome structure in higher plants is also unstable.”.
Line 61) Remove “unfortunately”.
Line 68) “…of cacti species,…” change to “…of a cactus species,…”.
Results
Lines 74 and 75) The mitogenome lengths in bp have a space in them that confuses the information. For example, 9, 026 bp, and 7, 040 bp should be written 9,026 bp and 7,040 bp with no internal spaces in the numbers. This same issue (spaces within numbers) occurs elsewhere in the manuscript, including figures, and should also be revised accordingly.
Line 77) “…Chain Reaction (PCR) experiment.” Change to “…chain reaction (PCR).”.
Line 81) “…Table S2 respectively…” change to “…Table S2, respectively.”.
Line 81) “The PCR products are…” change to “The PCR products were…”.
Line 83) “…here…” delete.
Figure 1 legend) Should be written in the present tense since it refers to the representation of results.
Line 100) “Next, we used long-reads…” change to “Long reads were subsequently used…”.
Lines 120-121) “…are potentially to mediate mitogenome recombination and to form putative multiple conformations.” This statement should be moved to the Discussion section since it is conjecture.
Line 129) “…plant and 9 variable genes…” this is confusing and should be explained further.
Table 1) The parentheses indicate the numbers of identified genes in the mitogenome. The reviewer suggests that the “x2” notation be shortened to simply “2”.
Line 150) “…are unknown sources…” change to “…are from unknown sources…”.
Line 152) “rrn26, and they all have two copies…” change to “…rrn26 all have two copies…”.
Line 158) “The gene…” has a This may be relatedsequence homology between the P. aculeata mitogenome and chloroplast genome, suggesting that these sequences were transferred between organelles during evolution.”.
Line 210) “The MTPT1 is the longest at 3, 962 bp.” Change to “MTPT1 is the longest at 3,962 bp.”.
Line 211) “..foreign…” suggest change to “…non-native…”.
Line 217) “…and so on…” delete.
Line 236) “…1…” change to “…one…”.
Line 276) “…its…” delete.
Lines 277-278) “It turns out that there are a lot of collinear blocks between them.” Change to “These species exhibit many common colinear mitogenomic sequences.”.
Line 278) “This may be related…” change to “Sequence colinearlity may be due…”.
Line 280) “…un-conserved…” reviewer believes that “…nonconserved…” is more appropriate.
Lines 281-284) The statements here are confusing to the reviewer and should be rewritten to be clearer.
Discussion
Line 291) “…It's…” change to “…it is…”.
Lines 300 and 302) “ … experiment…” delete.
Lines 302-303) “A total of five pairs of repetitive elements have been shown to mediate recombination in dandelion mitogenomes.” A citation is needed here to verify the presented information.
Lines 303-304) “Besides, the short repetitive elements are also confirmed to mediate genome recombination.” Change to “Short repetitive elements also confirmed such genome recombination.”.
Line 329) “Based…” suggest a new paragraph starting with this sentence.
Line 340) “In most plant lineages, the tRNA gene is lost after it is transferred from bacteria.” This statement needs a citation.
Materials and Methods
Line 404) “…we extract...” change to “…we extracted…”.
Line 411) “…we count those long-reads that completely span these query sequences,,,” change to “…we counted those long-reads that completely spanned these query sequences…”.
Line 418) “…is 25 µL…” change to “…was 25 µL…”.
Line 445) “…e-value is limited to less than 1e-05…” change to “…e-value was limited to less than 1e-05…”.
Line 463) “And then, a total of 31…” change to “A total of 31…”.
Conclusions
Lines 482-484) “There are two repetitive sequences in the P. aculeata mitogenome, which promote the rearrangement of the mitogenomes and generate multiple conformations, which was confirmed by long-reads and PCR experiments.” Change to “Long reads and PCR confirmed that there are two repetitive sequences in the P. aculeata mitogenome that promote the rearrangement of the mitogenomes and generate multiple motifs.”
The language and grammar in the paper were acceptable but included many inappropriate or colloquial passages that were pointed out to the authors in the reviewer report. After these errors are addressed, the quality of English is deemed to be acceptable.
Reviewer 2 Report
Review of Assembly of the complete mitochondrial genome of Pereskia aculeata revealed that two pairs of repetitive elements mediated the recombination of the genome by Feng et al. for the International Journal of Molecular Sciences
[I've also included this as a pdf in case this renders weird]
This is a strong and important contribution. The analyses in the paper seem to be sound but there are a few analyses that have been omitted that inform the process of recombination in plant mitochondrial genomes that need to be performed to make for a more complete description of repeats and their potential recombination activity. Also, there may be some minor clarification and more precision required in a few parts. I am recommending this for acceptance pending minor revision.
There is a lack of discussion of some of the most important prior studies of rearrangement and recombination in plant mitochondrial genomes in both the introduction and discussion. I would suggest the authors do some more literature review to better contextualize their findings including adding more discussion and appropriate citations – some of the avenues highlighted below are steps routinely done in some of the more important studies in this area and their omission is very curious.
Line 178 – Clarify “dominant inheritance” – if this is not referring to dominance in the Mendelian sense (it does not seem to be), please reword this. If it is, please elaborate.
Line 192 – What is the similarity (%) between the recombination-active repeats? High similarity could be indicative of ongoing gene conversion and concerted evolution. Whichever result is obtained for these two repeat sets, (high similarity suggesting frequent recombination, gene conversion, and concerted evolution, or low similarity suggesting the opposite) it should be discussed in the text. Similarity between repeats (NOT just e-values, which are contingent on database size and therefore far more difficult to interpret outside of the incredibly specific context of the BLAST searches used to identify repeats) for all repeats should be included in supplementary table S7.
Line 273-284 – This section is full of vague, qualitative statements. Some program, like GRIMM or a similar program, should be used to calculate the minimum number of genome rearrangements between each pair. For example, this would allow a discussion of rearrangement in terms of precise numbers of rearrangements rather than having vague statements gesturing toward rearrangement or genome turnover like “a lot” of collinear blocks or “relatively few” synteny blocks.
I look forward to seeing this manuscript published pending these minor, yet integral, revisions.

The English language quality is good - I suggest a round of editing and proofreading after my comments are considered and the manuscript is revised. Hopefully, any remaining minor errors will be caught in the copy-editing/proof stage.
